# Self-Healing Glass/Metakaolin-Based Geopolymer Composite Exposed to Molten Sodium Chloride and Potassium Chloride

**Patrick F. Keane** [1,*] [ID], **Rhys Jacob** [2] [ID], **Martin Belusko** [3] and **Frank Bruno** [1] [ID]

1   Future Industries Institute, University of South Australia, Mawson Lakes, SA 5095, Australia
2   Forschungszentrum Jülich GmbH, Structure and Function of Materials (IEK-2), 52428 Jülich, Germany
3   Mondial Advisory Pty Ltd., Hyde Park, SA 5061, Australia
*   Correspondence: patrick.keane@mymail.unisa.edu.au

**Abstract:** Geopolymers (GP) are a class of X-ray amorphous, nanoporous, nanoparticulate materials that can be mixed, poured, and cured under ambient conditions. Typically, geopolymers are made using a Group 1 (G1) alkali activator such as sodium or potassium metasilicate and an aluminosilicate precursor. An analogous material to GPs is ordinary Portland cement because of the similarities in processing, however, the resulting microstructure is more similar to that of a glass. Geopolymers are more thermally stable than OPC and can therefore be used in a variety of thermal energy storage systems, as energy storage is an increasing global concern. In this study, potassium metakaolin-based geopolymer composites containing glass particles and alumina platelets were manufactured, heated in air, and exposed to molten sodium chloride or potassium chloride under an air atmosphere. Results showed the formation of an amorphous self-healing geopolymer composite (ASH-G) that could contain molten G1 chlorides for over 200 h without signs of macro or microscopic chemical degradation. The filling of cracks by glass particles in the composite after heating to 850 °C makes this material self-healing. It was found that the morphology of ASH-G composites was more affected by temperature and duration than contact with corrosive molten chlorides in air. Future works include investigating the effect of molten salt on mechanical properties during initial heating, after prolonged heating, and the material compatibility with other molten Group 1 chloride eutectics.

**Keywords:** geopolymer; porosity; heat treatment; alumina; glass frit; chloride; molten salt

## 1. Introduction

Global energy consumption in 2021 was 595 exajoules according to statistical data provided by British Petroleum [1]. Of this energy consumed, 82% came from coal, natural gas, and oil, which equates to 36.3 gigatons of carbon dioxide released in 2021 alone [1,2]. Climate change caused by the release of greenhouse gases, as well as the high cost of fossil fuels, have made renewable energies and technologies an attractive option [1,3,4]. A major drawback of renewable energy sources is the inconsistent supply compared to the grid demand. Energy storage systems coupled with renewable energy would greatly lower the cost of energy as well as reduce the amount of $CO_2$ released.

There are several ways to store energy such as electrochemically, in the form of batteries, mechanically, in the form of pumped hydro, or thermally, in the form of hot or cold masses. Commonly deployed batteries (e.g., lithium-ion) require rare earth elements that are expensive and in limited supply [5], while hydroelectric batteries require natural or artificial height differences to take advantage of gravity, which limits locations where they can be placed [6,7]. Alternatively, thermal energy storage systems can be made from abundant materials and placed in a variety of locations [8].

In general, thermal energy storage (TES) can be classified into three broad categories: sensible, latent, and thermochemical. It can be further characterized based on temperature, with applications ranging from sub-zero (e.g., cool rooms), to low- to medium-temperature (e.g., HVAC), or high-temperature (e.g., concentrated solar power). Of the stated categories, sensible is the most often deployed owing to its relative simplicity [9].

In sensible TES, the energy stored relies on the total mass of the storage material, the specific heat of the storage material, and the temperature difference of the storage material. A major challenge of these systems is the footprint required to house the storage material, which in many low-temperature cases is water. To reduce the footprint of thermal energy storage systems, a higher temperature difference, and therefore a higher temperature material, can be used. At higher temperatures, molten salts such as nitrates, chlorides, or carbonates can be used owing to their favorable melting points (for latent heat applications) and low cost [10,11].

Of particular interest is the use of chloride materials as they are very abundant and low-cost [12]. For example, chlorine and sodium are the third and fourth, respectively, most abundant elements in Earth's oceans behind hydrogen and oxygen which makes sodium chloride salt abundant and relatively easy to process [13]. Sodium chloride, $NaCl$, is solid at ambient conditions and has a relatively high thermal conductivity, specific heat, and latent heat of fusion upon melting at 801 °C. These properties are ideal for storing thermal energy [14]. The major drawbacks of using $NaCl$ and other chlorides for thermal energy storage are corrosivity, chemical degradation, and operating temperatures. Molten salts have also been used to quench alloys, extract minerals from ore, and are found in potential nuclear reactors [15–17], making their containment an important research topic.

Container material for molten salts can be metal alloys, ceramics, or refractory composites. Common metal alloys such as stainless steel 304 and 316 have shown substantial levels of corrosion and require high-purity inert atmospheres [18–20]. Suitable alloys are not entirely corrosion-proof, use rare metals, and are expensive [21]. Ceramics require high amounts of time and energy by comparison [22]. Refractory bricks are affordable, and chemically and thermally stable but require joins and mortars which can be a major point of failure [23]. An alternative to these proposed solutions is geopolymer composites.

Geopolymers (GP) are a class of X-ray amorphous, nanoporous, nanoparticulate, inorganic covalently bonded material typically comprised of Group 1 (G1) aluminosilicates [24]. GPs are similar in processing to cements yet similar to glass in microstructure. Synthesis of GPs can be done at ambient temperatures and pressures by the addition and high shearing of aluminosilicate precursor, such as calcined kaolin (metakaolin), with an alkali activator solution. Acidic synthesis is an alternative method of producing similar phosphoric GPs, however, this route is not only relatively new compared to the alkali route but also utilizes phosphorous, which is much less common than elements used in the alkali synthesis route [25,26]. Prospect aluminosilicate precursors include various industrial waste ashes from metal processing, coal-fired power plants, and incinerators [25]. However, these tend to have high amounts of calcium, which is detrimental to geopolymer formation and maximum operating temperatures/environments [27]. This is due to the formation of calcium silicate hydrate phases, which are less thermally stable than covalently bonded aluminosilicates with minimal hydrates [27–29]. The repeatability of industrial waste ash composition is another concern, requiring attention to the individual batch formulation. Common alkali activator solutions are aqueous G1 hydroxides such as sodium and potassium and amorphous fumed silica to produce a solution of G1 metasilicate, which is also needed to fabricate geopolymers. Other alkali activating sources include G1 metasilicate, which is formed from the fusion of G1 carbonates with silica sand at high temperatures. Geopolymers made from G1 hydroxides have higher degrees of polymerization and therefore higher mechanical properties [30].

Metakaolin-based geopolymer composites can be made into complex shapes, easily incorporate functional phases, are thermally and chemically stable, and are relatively inexpensive. Additionally, ceramics can be produced by heating geopolymers to 950 °C [31].

For these reasons, metakaolin-based geopolymers could be a potentially suitable containment material for molten chlorides [32,33]. Two obstacles that need to be overcome are microcracking and significant shrinkage/warping of the geopolymer matrix after heating to 800 °C [34,35]. One method to prevent significant shrinkage during initial heating is the addition of alumina platelets [36]. The homogenous dispersion of glass-forming material in the composite has also been shown to be a successful strategy for filling microcracks [37,38]. The filling of microcracks by a uniformly distributed phase in the composite, upon heating, makes the composite self-healing [31].

Previous work on this conducted by Keane et al. (2022) showed that heat-treated, potassium metakaolin geopolymer containing homogeneously mixed alumina platelets and System 96 © glass particulates of size 10–250 μm did not chemically interact among themselves and produced a self-healing composite with uniform glaze [31]. X-ray diffraction analysis of bulk powder samples showed no new crystal formation among the phases after heating to 900 °C.

Therefore, the purpose of the current study is to investigate the compatibility and performance of a self-healing geopolymer composite exposed to either molten sodium chloride or potassium chloride to determine if any chemical degradation occurs and if this material can be used to contain molten salt at high temperatures. Potassium chloride was selected because of its similarity to sodium chloride, as well as the presence of potassium in the geopolymer composite. The aim of this research is to successfully contain molten chlorides for extended durations. If successful, the containment of molten salts by geopolymers can be used as an inexpensive solution to energy storage and fossil fuel use reduction [11].

## 2. Methodology

### 2.1. Geopolymer Composite Preparation

A metakaolin-based (MK) potassium geopolymer matrix (KGP) was prepared by the addition of MetaMAX $^®$ from BASF and potassium metasilicate solution, also known as waterglass (KWG). MetaMAX $^®$ is an aluminosilicate kaolin clay that has been heated to approximately 750 °C and has an average particle size of 1.3 μm [39,40]. A potassium metasilicate solution was synthesized by dissolving 85% assay potassium hydroxide flake and fumed silica, from ChemSupply Australia, into deionized water. The mixture was stirred at 300 rpm for 24 h using a magnetic stir bar, ensuring the total dissolution of the fumed silica. Metakaolin clay was then high-sheared into the KWG solution for 5 min at 2000 rpm using an IKA $^®$ RW 20 high-shear mixer and dissolver attachment to produce a homogenous KGP slurry. The KGP slurry was de-aerated for 5 min using a vibration table. Equation (1) and Equation (2) show the formation of KWG and KGP.

$$2KOH + 2SiO_2 + 10H_2O \ \rightarrow K_2O \bullet 2SiO_2 \bullet 11H_2O \tag{1}$$

Equation (1): Formation of potassium metasilicate solution from $SiO_2$, deionized water, and KOH

$$[K_2O + 2SiO_2 + 11H_2O] + [Al_2O_3 \bullet 2SiO_2] \rightarrow [K_2O \bullet Al_2O_3 \bullet 4SiO_2 \bullet 11H_2O] \tag{2}$$

Equation (2): Formation of potassium geopolymer from metakaolin clay and potassium metasilicate solution

Fifty-micron alumina platelets from Micro Abrasives were then added to the KGP slurry and high-sheared for 5 min followed by 2 min on a vibration table to remove entrapped air. System 96 $^®$ glass frit powder of particle size 10–250 μm from Oceanside Glass and Tile was then added to the mixture, high-sheared for 5 min, and de-aerated for another 5 min. The homogenous mixtures were then poured into plastic molds and placed in a 100% relative humidity curing chamber for five days. Plastic molds were made from modified PVC plumbing pipe and caps, as well as PVC rod. Weight percents, both reference and measured densities, and calculated volume percents of each phase are presented in Table 1. The elemental composition of glass frit powder was measured using EDS and is

presented in Table 2. The glass frit powder is most similar to sodalime glass with the minor addition of impurities, which are used to prevent crystallization upon melting and cooling among other properties [41].

**Table 1.** Composition of Alumina-Glass-Geopolymer Composites.

| Phase | Weight (%) | Theoretical Density (g/cm$^3$) | Measured Density (g/cm$^3$) | Calculated Volume (%) |
|---|---|---|---|---|
| KGP | 50 | 2.0481 [24] | 1.86 | 57.8 |
| Glass Frit | 35 | 2.5 [42] | 2.47 | 33.2 |
| Alumina Platelet | 15 | 3.97 [43] | 3.88 | 9.0 |

**Table 2.** Elemental Composition of Glass Frit Powder (10–250 μm).

| Element | Weight (%) |
|---|---|
| O | 44.0 |
| Na | 12.5 |
| Mg | 0.8 |
| Al | 1.6 |
| Si | 33.1 |
| K | 0.7 |
| Ca | 4.8 |
| Zn | 1.9 |
| Y | 0.7 |
| Total | 100.0 |

## 2.2. Heating/Sintering

Cured composites were demolded and acclimated to ambient humidity for 48 h before being placed into an air furnace. Composites were then heated to 900 °C for 5 h using heating and cooling rates of 2.5 or 10 °C/minute, respectively. The target temperature and ramp rate were selected as successful samples were previously produced using 900 °C and 2.5 °C/min respectively [31]. The ramp rate of 10 °C/min was selected it is a conventional heating rate for sintering ceramics [44]. Next, heat-treated geopolymer composites were filled with a known amount of sodium chloride or potassium chloride.

Isothermal container testing was conducted in an air muffle furnace for 240 h using initial heating and final cooling rates of 10 °C/min. For each Group 1 chloride, one empty control composite was placed next to a composite containing G1 chloride resulting in four composites total (two control composites [S3,5], one composite containing NaCl [S2], and one composite containing KCl [S4]). NaCl testing was conducted at 805 °C and KCl testing was conducted at 775 °C. These temperatures were selected to minimize mass loss but ensure melting.

## 2.3. Group 1 Chloride Preparation

Sodium chloride and potassium chloride were purchased from Sigma Aldrich. Both chlorides were ACS grade with a purity of >99%. Approximately 12 g of sodium chloride or potassium chloride were added to cylindrical geopolymer composite crucibles that had been heated to 900 °C. Crucibles filled with salt were then heated to 5 °C above their respective melting points and held for 240 h to indicate if a chemical reaction or degradation occurred.

### 2.4. Mass/Length Change, Density, and Apparent Porosity

The weight, height, and diameter of the geopolymer composites were measured using a mass balance accurate to 0.01 g and calipers accurate to 0.01 mm. Measurements were taken before and after heating, as well as after exposure to the molten Group 1 chlorides. Mass losses were determined by comparing sample weights before and after heating. Length changes were determined by comparing sample outer diameters before and after heating, similar to previous work [31]. Heat-treated sample densities and open porosities were measured before and after exposure to molten salt using the Archimedes method outlined in ASTM C20. Deionized water at 24 °C was used as the reference fluid. Samples were dried at 130 °C for 24 h and weighed, $W_{Dry}$. Next, samples were submerged in boiling water for at least 2 h to fill all open pores with fluid. Once cooled and still submerged, suspended samples were weighed to offset buoyant force, $W_{Sub}$. Finally, saturated samples were wiped of free water and weighed, $W_{Sat}$. Equations (3)–(7) below show how to calculate open porosity and bulk density using described sample weights.

| | |
|---|---|
| $V_{Bulk} = \frac{(W_{Sat} - W_{Sub})}{\rho_{Water}}$ | Equation (3). Bulk Volume |
| $V_{Open\ Pores} = \frac{(W_{Sat} - W_{Dry})}{\rho_{Water}}$ | Equation (4). Open Pore Volume |
| $V_{Matrix} = \frac{(W_{Dry} - W_{Sub})}{\rho_{Water}}$ | Equation (5). Apparent Volume |
| $\rho_{Bulk} = \frac{W_{Dry}}{V_{Bulk}} = \frac{W_{Dry} * \rho_{Water}}{(W_{Sat} - W_{Sub})}$ | Equation (6). Bulk Density |
| $Open\ Porosity = \frac{V_{Open\ Pores}}{V_{Bulk}} = \frac{W_{Sat} - W_{Dry}}{W_{Sat} - W_{Sub}}$ | Equation (7). Open Porosity |

### 2.5. Microcharacterization

Samples were polished on a Struers Tegra grinding system using silicon carbide paper of 180 and then 1200 grit. Further polishing was done on a Presi Mecatech 250 polishing system down to 6 μm. A carbon coater deposited a conductive layer of approximately 20 nm thickness to prevent electron buildup during SEM/EDS analysis. Micrographs were collected using an Olympus SC50 optical microscope, a Zeiss Merlin field emission electron gun SEM, and silicon drift detector EDS. Fracture surfaces and polished surfaces were used to observe porosity, localized heterogeneity, and chloride presence. ImageJ software was used in post-processing to determine the size of features.

X-Ray diffraction patterns of samples were measured using a Panalytical Empyrean X-ray diffractometer using a step size of 0.02° from 10 to 90 2θ. The X-ray source was CuKα with a nickel filter monochromator and an approximate wavelength of 1.54 Å. Recovered salt samples were ground to less than 50 μm particulates. Sample diffraction patterns were then compared to the International Centre for Diffraction Data PDF4+ database for confirmation.

## 3. Results & Discussions

Geopolymer composites containing fifty-micron alumina platelets and glass particulates as seen in Figure 1 were manufactured, cured for seven days, and heated to 900 °C for five hours. The high angularity of these particles allows for a relatively higher average free space among particles [45]. This higher average free space among particles allows for the molten glassy phase to flow throughout the composite more easily at elevated temperatures once the matrix phase has uniformly shrunk [46]. This may seem counterintuitive as shrinking causes a reduction of space, however, the alumina phase remains rigid [36]. It can be theorized that capillary action and/or liquid phase sintering of the shrinking matrix phases drew in the softened, highly viscous glass particulates upon further heating [47–49].

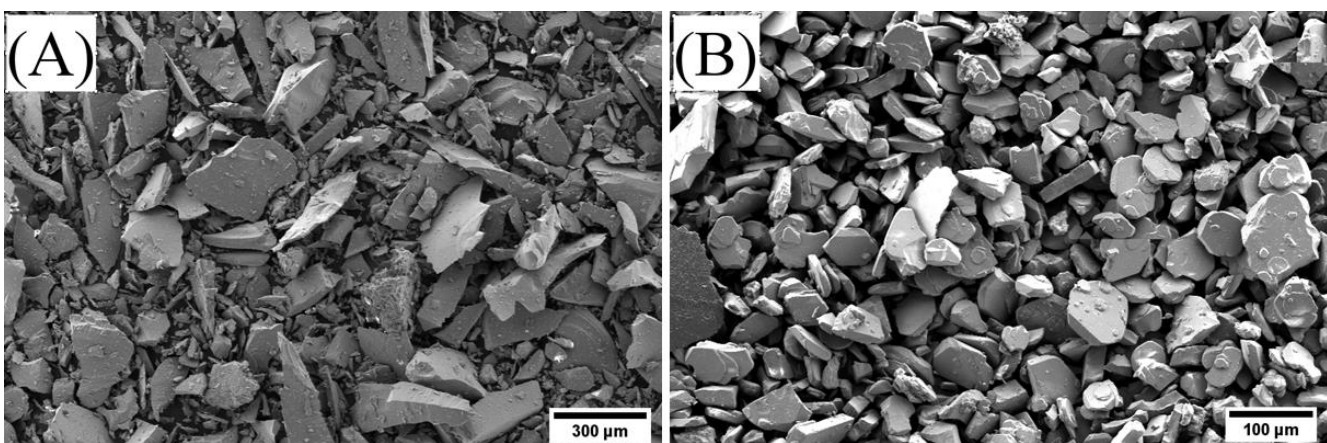

**Figure 1.** Glass frit powder as received (**A**). Alumina platelets (50 µm) as received (**B**).

Figure 2 below shows crucibles before and after heating using slow (2.5 °C/min) and fast (10 °C/min) heating rates. A uniform high sheen glaze was observed among geopolymer composites that were heated at 2.5 °C/min to 900 °C for five hours; however, samples heated at 10 °C/min were nonuniformly deformed and did not form a uniform glaze. The increased volume in Figure 2B indicates an increase in porosity from the initial sample, which may have merit in other applications. In this study, however, uniform volume change after heating is a preliminary requirement for a suitable molten salt containment material. Therefore, only composites with a heating rate of 2.5 °C/min were used for salt containment testing.

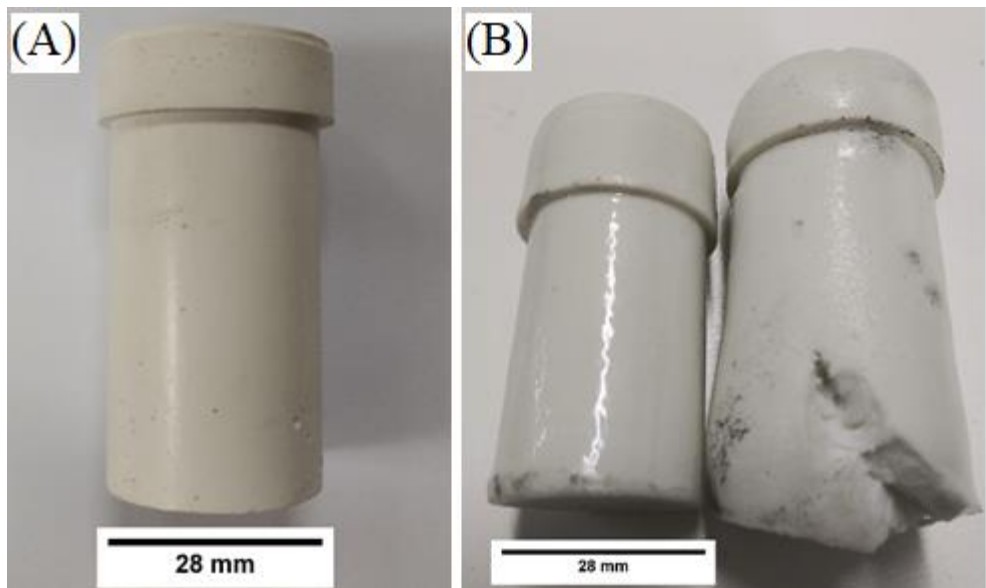

**Figure 2.** ASH-G composites before (**A**) and after (**B**) heating to 900 °C for five hours using 2.5 °C/min (**left**) and 10 °C/min (**right**) heating and cooling rates.

### 3.1. Effect of Initial Heat Treatment

A linear shrinkage of 7.7% and mass loss of 12.0% occurred after heating cured ASH-G composites to 900 °C for five hours using heating/cooling rates of 2.5 °C/min. Additionally, a uniform glaze of approximately 200 µm thickness formed on the 2.5 °C/min heat-treated polished samples as seen in Figure 3. The densities before and after initial heating to 900 °C were 1.93 g/cc and 2.32 g/cc, respectively. Open porosities before and after initial heating to 900 °C were 19.95% and 0.73%, respectively. The results are shown in Table 3.

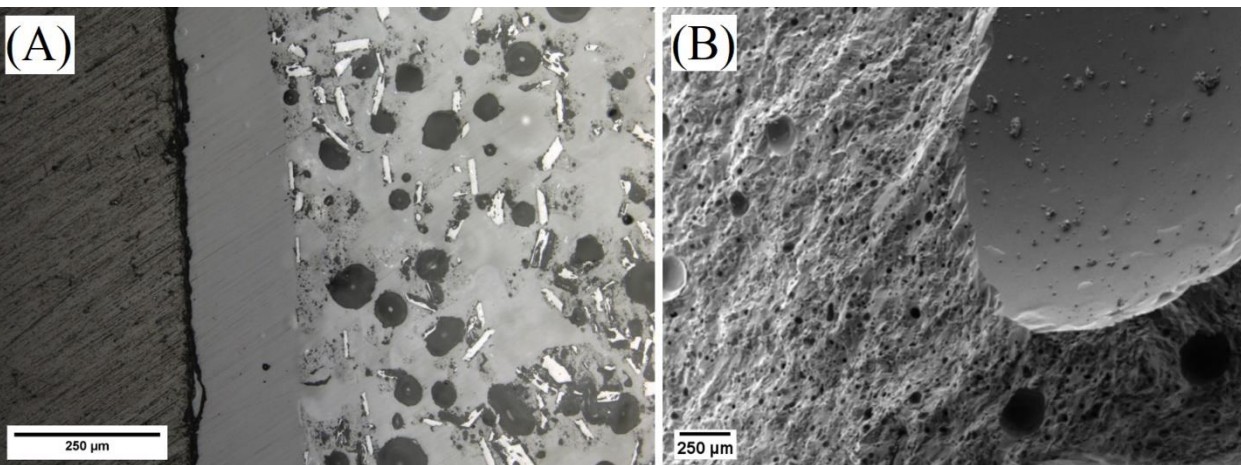

**Figure 3.** Micrographs of polished (**A**) and fractured (**B**) ASH-G composites after heating to 900 °C for 5 h using heating/cooling rates of 2.5 °C/min.

**Table 3.** Physical Changes Due to Initial Heating to 900 °C.

| Sample | Linear Shrinkage (%) | Mass Loss (%) | Density (g/cc) | Open Porosity (%) |
|---|---|---|---|---|
| Room Temperature | 0 | 0 | 1.93 | 19.95 |
| Post 2.5 °C/min Heat Treatment | 7.7 | 12.0 | 2.32 | 0.73 |

Previous work indicated no chemical reaction occurred among the three phases according to XRD and EDS results [31]. After heating plain KGP without additives to 900 °C a series of crystalline peaks appeared in an XRD analysis [31]. Microcracks formed during geopolymer matrix dehydration/densification were filled by molten glassy phase as indicated by little-to-no open porosity in the heated samples as expected [31,50]. Figure 4 shows how the glass phase filled much larger cracks after heat treatment. This removed the presence of discrete glass particulates and introduced oblong, form-fitting regions of glass and geopolymer matrix [31]. This phenomenon makes this material a good candidate for the containment of high-temperature liquids as the liquid cannot penetrate the glassy layer, leaving the structural material untouched and unaffected.

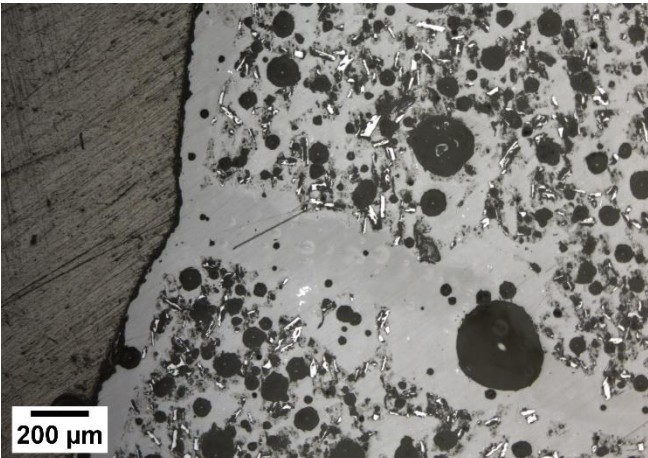

**Figure 4.** Micrograph of polished ASH-G composites after heating to 900 °C for 5 h using heating/cooling rates of 2.5 °C/min.

### 3.2. Effect of Molten Salt Exposure/Prolonged Heating

Once the composites were heated and a uniform glaze was formed, the samples were cooled, and weights measured. A known amount of either 99.9% assay sodium or potassium chloride was added to the samples and an alumina cover was placed over the opening to minimize contamination and mass loss as seen in Figure 5.

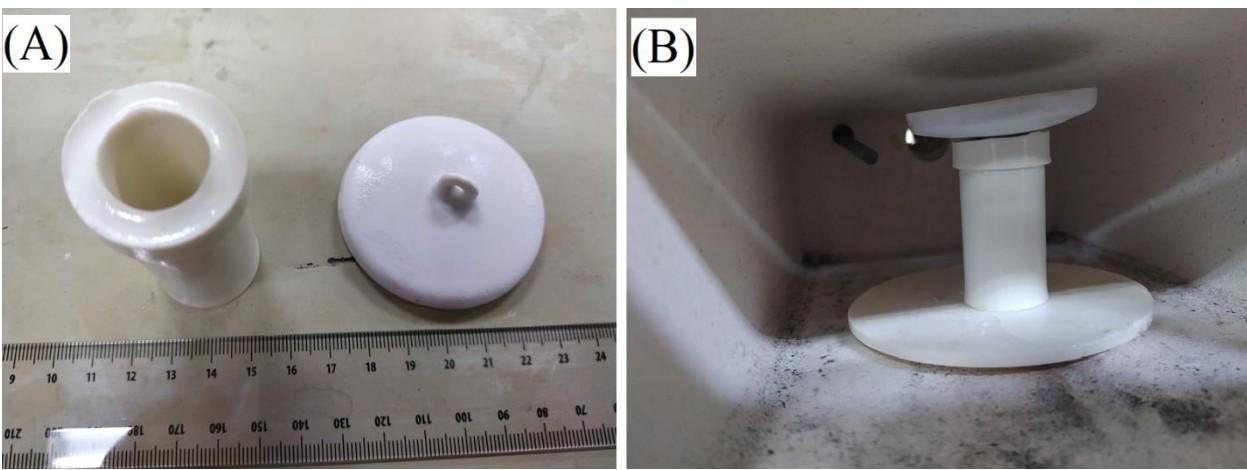

**Figure 5.** ASH-G composites before (**A**) and after (**B**) filling with Group 1 chloride salts.

Amorphous self-healing geopolymer containers were heated with and without Group 1 chloride to ~ 5 °C above their melting points—805 °C and 775 °C for sodium chloride and potassium chloride, respectively. Both test and control samples were held at these temperatures for 240 h. Once cooled, the ASH-G samples were measured for changes in mass and length. Solidified sodium chloride and potassium chloride samples were also taken for investigation with X-ray diffraction. ASH-G samples were then ultrasonicated in DI water for 15 min to dissolve any residual salt on the surface. Density and open porosity measurements were collected according to ASTM C20 using 24 °C de-ionized water as reference material. The results can be seen in Table 4.

**Table 4.** Physical Changes Due to Prolonged Exposure to Molten G1 Chlorides.

| Sample | Sample Number | Linear Shrinkage (%) | Mass Loss from Cured GP (%) | Density (g/cc) | Open Porosity (%) | Salt Mass Loss (%) |
|---|---|---|---|---|---|---|
| Room Temperature | 1 | - | - | 1.93 | 19.95 | - |
| Post Heat Treatment | 1 | 7.7 | 12.0 | 2.32 | 0.7 | - |
| Post NaCl Exposure (805 °C) | 2 | 8.6 | 11.9 | 2.37 | 2.3 | 4.1 |
| Post NaCl-test Control (805 °C) | 3 | 8.6 | 12.0 | 2.38 | 2.1 | - |
| Post KCl Exposure (775 °C) | 4 | 8.2 | 11.9 | 2.32 | 1.4 | 2.4 |
| Post KCl-test Control (775 °C) | 5 | 8.2 | 11.9 | 2.33 | 1.5 | - |

Data indicates that the presence of molten salt had little-to-no effect, while temperature and duration had a more significant effect. After the mass loss during initial heating, no samples experienced further mass loss due to prolonged heating. Geopolymer composites that were heated to 805 °C, containing molten sodium chloride, showed higher linear shrinkage and did not retain a surface glaze, as seen in Figure 6. In comparison, geopolymer

composites heated to 775 °C, containing molten potassium chloride, did retain a reduced surface glaze of approximately 180 μm, as seen in Figure 7, and showed slightly less linear shrinkage compared to the 805 °C samples. The presence of glaze on composite containing potassium chloride in Figure 7 could also be attributed to potassium chloride being the most stable of the Group 1 chlorides, and least likely to react, according to the enthalpies of formation as well as slightly lower dwell temperature [51].

It can be hypothesized that the lack, or reduction, of the glaze, is due to some glassy phase material migrating into the bulk of the composite during reheating above the glass softening point of 573 °C, according to previous work [31]. EDS analysis of geopolymer composites in Figures 6 and 7 show the lack, or presence, of surface glaze via aluminum concentration. Areas of brighter purple indicate higher aluminum content and dimmer areas indicate lower aluminum content. Figure 8 shows lighter discrete alumina while connected irregular medium or darker phases are potassium geopolymer and glassy phase, respectively. No microcracks can be seen in the bulk of samples exposed to molten salt, indicating no chemical degradation occurred to the containment material on a microscopic level.

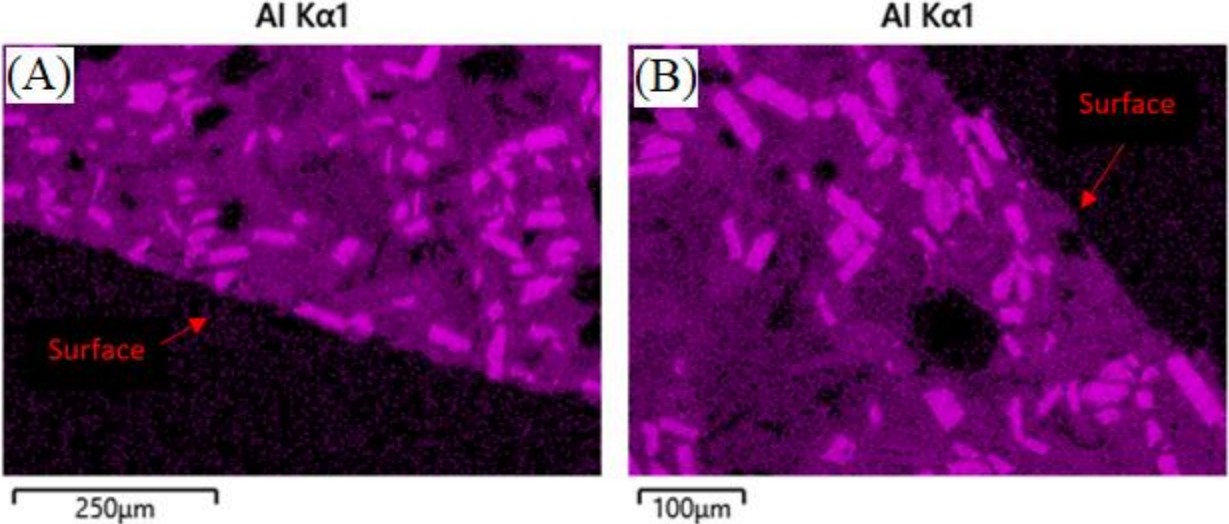

**Figure 6.** Surfaces of ASH-G composite after heating to 805 °C for 240 h without (**A**) and with (**B**) NaCl. No glaze is present in either case.

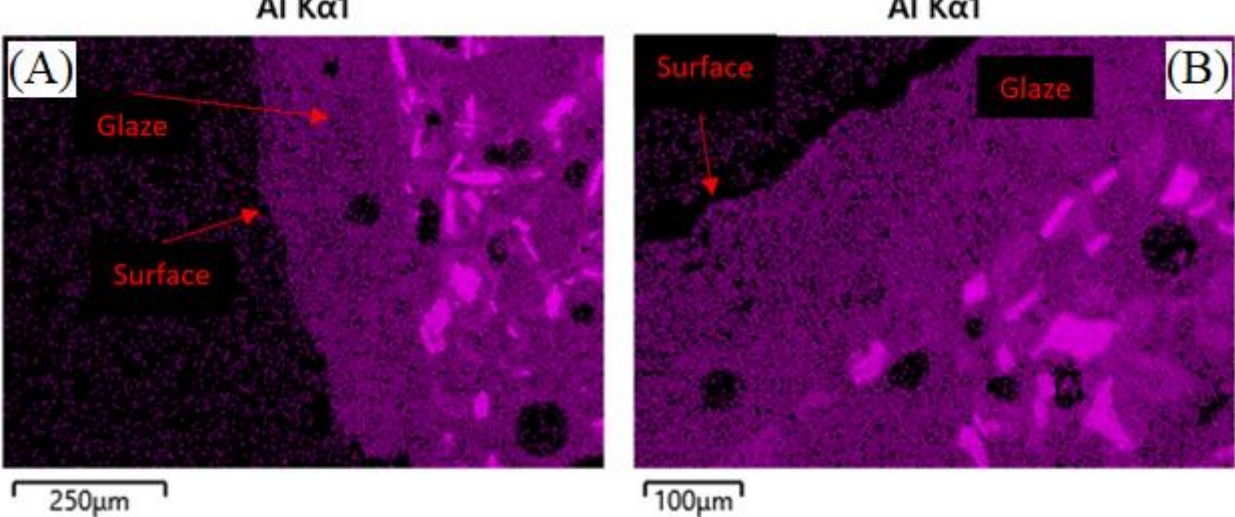

**Figure 7.** Surfaces of ASH-G composite after heating to 775 °C for 240 h without (**A**) and with (**B**) KCl. A reduced glaze is present in both cases.

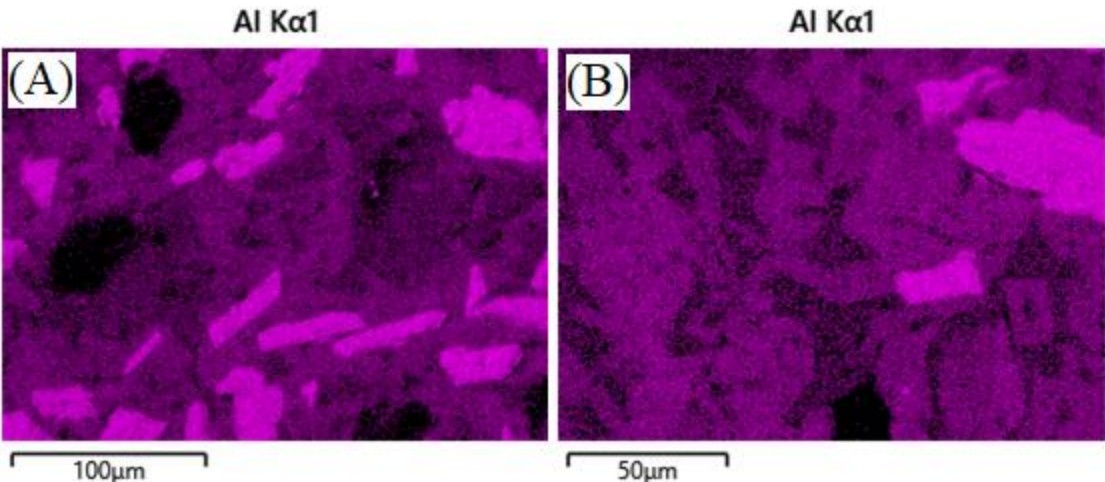

**Figure 8.** Bulk ASH-G composite after heating to 805 °C for 240 h with NaCl (**A**) and heating to 775 °C for 240 h with KCl (**B**). No microcracks are present.

To further check for the chemical reaction between chloride salts and geopolymer composites, bulk powder XRD analysis of ground salt samples was conducted. Representative salt samples were taken from the as-received supply and post-test crucibles. The salts were dried at 120 °C for 24 h and ground to <50 μm. The results can be seen in Figure 9 and show the presence of potassium chloride in sodium chloride samples and sodium chloride in potassium chloride samples.

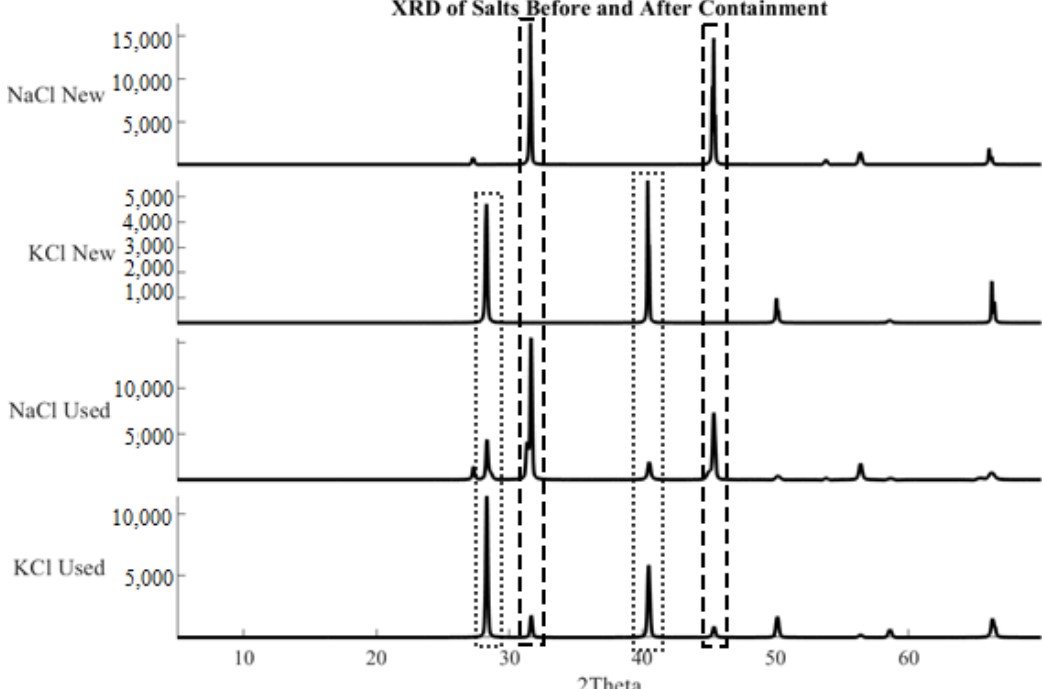

**Figure 9.** X-ray diffraction patterns of NaCl and KCl before and after prolonged heating in ASH-G composites. Salt samples after prolonged exposure to ASH-G show the presence of both Group 1 chlorides.

The exchange of Group 1 elements can be attributed to diffusion, and not dissolution, of the container material. Dissolution of the container material would result in microcracks, which are not present in the EDS micrographs. No other crystalline phases were observed in the recovered salt samples. If chemical degradation occurred, alumina platelets would

be free to dislodge and appear in the recovered salt samples. The results indicated that the geopolymer composites did not react with air, molten salt, or chemical species generated from molten salt-air interactions to form crystalline phases.

The ability of geopolymers to incorporate additive phases of different shapes, sizes, quantities, and chemistries into stable composites is valuable and needs to be explored. There is a range of solutions for the containment of different molten salts in different geometries for different applications is a testament to the utility of engineered geopolymer composites. While no microscopic or macroscopic chemical degradation was found, how the flexural strength of geopolymer composites exposed to both molten salt and temperature changes is of particular interest. Previous work done by Tao Ai et al. found that potassium metakaolin-based geopolymer without additives exhibited superior densification and compressive strengths when heated in molten salt rather than air [46]. Initial heat treatment to 300 °C in air followed by further sintering in molten salt could not only produce stronger, more dense composites but also eliminate the contamination of molten salt over extended periods.

### 4. Conclusions

In this study, potassium metakaolin-based geopolymer composites containing alumina platelets and glass frit powder were formed into cylindrical crucibles. Composites were then heated to 900 °C in air for five hours using slow (2.5 °C/min) and fast (10 °C/min) heating/cooling rates. It was found that slow rates of heating and cooling caused samples to have an increased density, loss of nearly all open porosity, and develop a uniform glassy glaze of approximately 200 μm. This material can be called an amorphous self-healing geopolymer composite (ASH-G) as microcracks that occurred during heating were filled with glass phase. It is hypothesized that cracked samples can be slowly re-heated to high temperatures and the cracks healed with the molten glassy phase. Composites heated and cooled using 10 °C/min did not densify, did not produce a surface glaze, and deformed irregularly. It is speculated that sintering failure occurred during heating rather than cooling.

The as-produced amorphous self-healing geopolymer composites were able to contain molten Group 1 chlorides for 240 h. Significant findings include:

- The density and open porosity of the ASH-G samples increased and decreased, respectively.
- Molten sodium chloride and potassium chloride did not chemically react with the ASH-G container after 240 h
- A total of 4 wt% of molten salt was lost after 240 days in air
- ASH-G samples were more affected by temperature than exposure to molten salt

Analysis of the sodium and potassium chlorides indicates an ion exchange of potassium and sodium, respectively. It is shown that this occurred due to diffusion rather than the dissolution of the container material. The dissolution of any phase of the container material would result in the presence of inert crystalline alumina in the recovered salt. Future work includes mechanical evaluation to determine embrittlement or reduced flexural strength, as well as exposure to sodium/potassium chloride eutectic ($T_{melt}$ = 665 °C) to determine salt degradation and thermocycling.

**Author Contributions:** Conceptualization, P.F.K., R.J., M.B. and F.B.; Methodology, P.F.K.; Formal analysis, P.F.K. and R.J.; Investigation, P.F.K.; Resources, F.B.; Data curation, P.F.K. and R.J.; Writing – original draft, P.F.K.; Writing – review & editing, P.F.K., R.J., M.B. and F.B.; Supervision, R.J., M.B. and F.B.; Project administration, M.B. and F.B.; Funding acquisition, F.B. All authors have read and agreed to the published version of the manuscript.

**Funding:** This research was funded by The Australian Government Research Training Development grant, The Alexander von Humboldt Foundation Early Career Post Doctorate Grant, and The Australian Solar Thermal Research Initiative P22 Grant. The APC was provided by Applied Sciences.

**Data Availability Statement:** The data that support the findings of this study are available from the corresponding author, P.F.K., upon reasonable request.

**Acknowledgments:** The authors acknowledge Microscopy Australia for the use of facilities at the Future Industries Institute at the University of South Australia. The authors thank Scott Chemical Pty Ltd. for providing the metakaolin used in this study. Rhys Jacob would gratefully like to acknowledge the Alexander von Humboldt Foundation for providing funding to support this work.

**Conflicts of Interest:** The authors declare no conflict of interest.

## Abbreviations

ASH-G—Amorphous Self-Healing Geopolymer; DSC—Differential Scanning Calorimetry; EDS— Energy Dispersive Spectroscopy; G1—Group 1; KGP—Potassium Geopolymer; KWG—Potassium Waterglass; MK—Metakaolin; SEM- Scanning Electron Microscope; TES—Thermal Energy Storage; XRD—X-ray Diffraction.

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
