# Peer review of "Self-Healing Glass/Metakaolin-Based Geopolymer Composite Exposed to Molten Sodium Chloride and Potassium Chloride"

_applsci, doi:10.3390/app13042615_

Round 1
Reviewer 1 Report (New Reviewer)
The manuscript titled ” Amorphous Self-Healing Geopolymer Composite Exposed to Molten Sodium Chloride and Potassium Chloride” is meaningful and interesting. There are few research focusiong on geopolymer exposed to molten salts. After following revision paper can be acceppted.
1.The title “Amorphous Self-Healing Geopolymer Composite Exposed to Molten Sodium Chloride and Potassium Chloride” should be modified as the Lack of limitation for geopolymer composites such as “Self-Healing Glass/metakaolin-based Geopolymer Composite Exposed to Molten Sodium Chloride and Potassium Chloride”
2.Previsous works showed that NaCl can be directly added into the geopolymer slurry, which maybe useful for introduction parts for the backgroud and futher study. “Facile synthesis of porous geopolymers via the addition of a water-soluble pore forming agent”.
3. In the introduction parts, geopolymer composites and alkali activated composites should be added such as: ZHANG, Xiaohong, et al. Porous geopolymer composites: A review. Composites Part A: Applied Science and Manufacturing, 2021, 150: 106629. Humur, Ghassan, and Abdulkadir Çevik. "Mechanical characterization of lightweight engineered geopolymer composites exposed to elevated temperatures." Ceramics International 48.10 (2022): 13634-13650. And the introduction about Self-Healing geopolymers is not enough as well.
4. The detail how to make the tube-type composites should be given.
5. Fig. 1 can be modified the size can be uniformed, and the labels(a)(b) should be added rather than left right. The two figures can be list in a row.
6. The size for the lables(A)(B) should be uniformed in Fig.2 and Fig. 3 and Fig.4. And Fig.2 and Fig.3 can be merged. The shooting point of viewfor Fig.3 is different it is better to show in same shooting angel.
7. Fig.4(B) is which part in the Fig.4(A)?
8. It is better to give the XRD data for the composites before and after sintering.
7. The labels(a)(b) should be added rather than left right in Fig. 5 and Fig.6. And Fig.5 and Fig.6 can be merged.
8. Fig.7 and Fig.8 can be merged.
9. The main characteristic of the composites should be compared with previous works.
10. There are many of refereces come from other route rather than journals papers. It is seems that authors lack of the kowledge for geopolymer composites for high temperatuer expose.
11. “Error! Reference source not found.. I” And various colors showed. In any way, lots of Errors have been showed, authors should pay more attention for the format before submited.
Author Response
Reviewer's suggestions were incorporated in the latest version of the manuscript where applicable. These include highlighted portions.

Reviewer 2 Report (New Reviewer)
Manuscript ID: applsci-2049641
Title: Amorphous Self-Healing Geopolymer Composite Exposed to Molten Sodium Chloride and Potassium Chloride
Comments to authors:
This paper reports the performance of amorphous self-healing geopolymer composite exposed to either molten sodium chloride or potassium chloride under air atmosphere. This is well-written manuscript with acceptable novelty. However, to improve the quality of the manuscript, please address the following comments for a major revision:
1) Information about the significance and problem statement should be highlighted before the results section in the Abstract.
2) There is a need to improve the conclusions/results in the abstract.
3) The novelty and significance of the present work should be highlighted in the last paragraph of the Introduction section. The objective/aim of the provided research needs to be clearly defined in the introduction section.
4) The authors are recommended to add more latest relevant literature review on such works.
5) The methodology is well defined however the authors are recommended to add more detailed description in pictorial form about the methodology of the provided research.
6) In the methodology section it is stated that “cured composites were heated to 900 °C for five hours using heating and cooling rates of 2.5 or 10 °C/minute.” Please state the reason of selecting the heated temperature, duration and rate for the composites.
7) Please add clear information for test samples.
8) Please recheck the figures citation in the text. I think there is some sort of error in the text.
9) The results/Discussion is the most vital part of any research paper. This section needs to be rewritten by providing the comparison of the results with the latest literature. Please take into consideration some latest similar studies.
10) The ‘’Conclusion” section is simply summary of the results. There is a lack of analysis and interpretation. This section needs to be rewritten. Conclusions should be refined and briefly presented.
11) Future recommendations can be added.

Author Response
Reviewer's suggestions were incorporated in the latest version of the manuscript where applicable. These include highlighted portions.

Reviewer 3 Report (New Reviewer)
In this study, potassium metakaolin-based geopolymer composites containing glass particles and alumina platelets were manufactured, heated in air, and exposed to molten sodium chloride or potassium chloride under an air atmosphere. The research is interesting, but there are still many problems that need to be further corrected. The problems are as follows.
1. In the introduction, this sentence ‘However, these tend to have high amounts of calcium which is detrimental to geopolymer formation and maximum operating temperatures/environments’ seems to be problematic. As far as I know, the calcium is conducive to the formation of geopolymer strength, rather than a negative effect. Please check carefully.
2. In the introduction, in addition to alkali-activated geopolymers, there are acid-activated geopolymers. These acid-activated geopolymers still have good mechanical properties. Some recent related literature should be added in the paper to make the paper more interesting. e.g.
① A novel acidic phosphoric-based geopolymer binder for lead solidification/stabilization.
④ Environmental behavior and engineering performance of self-developed silico-aluminophosphate geopolymer binder stabilized lead contaminated soil.
3. In the Methodology section, the chemical composition, mineral composition and grain grading curve of the used material should be given. These contents are very important for the subsequent analysis. Please add .
4. The following problematic sentences appear in many places of the text,e.g. Weight percents of each phase can be seen in Error! Reference source not found. Please check it carefully and correct it.
5. In Table 1, the density parameters of three chemical materials are derived from three different citations. In fact, you need to measure these three indicators yourself.
6. In page 7, the first paragraph of Effect of Molten Salt Exposure/Prolonged Health should belong to the methodology section and should be moved to the front. The results and discussion section should only present the results and discussion. Please make changes.
7. In Figure 9, it can not be seen in the XRD spectrum which samples are heated before and after the XRD, and there should also be XRD of the control group, and then in-depth analysis. Moreover, in the XRD shown in Figure 9, the chemical substance corresponding to the peak in each XRD diagram should be given. Please revised it.
8. The conclusion needs to be rewritten. The conclusion is too long, and you should simplify it. The conclusion should include important findings in the article. Some content can be written in the discussion section. Please revised it.
Author Response
Reviewer's suggestions were incorporated in the latest version of the manuscript where applicable. These include highlighted portions.

Round 2
Reviewer 1 Report (New Reviewer)
Figs. 5-8 for the EDS result can be merged.
Author Response
Please see attached with comments. Figures have been corrected
Reviewer 3 Report (New Reviewer)
The author's revision is perfunctory and did not strictly follow my previous comment. Especially for the material section, Table 1 does not belong to chemical components, and Table 2 is also the content of chemical components. The author may not understand material science. The chemical components are obtained by XRF, the mineral components are determined by XRD, and the particle size is determined by particle size analyzer. In general, the author's revision is perfunctory and I did not see the corresponding response. Please make the corresponding revision and Response carefully according to the first comment.
Author Response
Please see attached for responses

This manuscript is a resubmission of an earlier submission. The following is a list of the peer review reports and author responses from that submission.
Round 1
Reviewer 1 Report
The authors have reported a timely research. However, the research methodology and the results and discussion section lacks significant merit.
The interpretation of the results is vague and is not supported by experimental data.
Here are some examples:
- The title, work scope, and discussion section emphasize on 'self-healing'. I have not found any evidence of 'self-healing'. The authors must proof the self-healing by experiment rather than vague conjecture.
- The authors suggested no significant mass loss. The experimental design for mass loss is flawed.
- There is no reference sample to compare the degradation/stability aspect of the samples. Same is true for the 'self-healing' claim.
Reviewer 2 Report
Unfortunately, the work does not provide any self-healing experiments, self-healing information for the readers, and discussion with the recent literature about the topic. I do not recommend to publish the paper.